# Graph-theoretical formulation of the generalized epitope-based vaccine design problem

Emilio Dorigatti [1,2]*, Benjamin Schubert [2,3]

**1** Department of Statistics, Ludwig Maximilian Universität, München, Germany, **2** Institute of Computational Biology, Helmholtz Zentrum München – German Research Center for Environmental Health, Neuherberg, Germany, **3** Department of Mathematics, Technical University of Munich, Garching bei München, Germany

\* edo@stat.uni-muenchen.de

**Data Availability Statement:** The source code and all datasets analysed in this study are available at https://github.com/SchubertLab/GeneralizedEvDesign.

## Abstract

Epitope-based vaccines have revolutionized vaccine research in the last decades. Due to their complex nature, bioinformatics plays a pivotal role in their development. However, existing algorithms address only specific parts of the design process or are unable to provide formal guarantees on the quality of the solution. We present a unifying formalism of the general epitope vaccine design problem that tackles all phases of the design process simultaneously and combines all prevalent design principles. We then demonstrate how to formulate the developed formalism as an integer linear program, which guarantees optimality of the designs. This makes it possible to explore new regions of the vaccine design space, analyze the trade-offs between the design phases, and balance the many requirements of vaccines.

## Author summary

Diseases such as Cancer, AIDS, Hepatitis C, and Malaria, infect and kill millions of people every year. In spite of all our efforts, a cure for those diseases remains elusive. Among all possible approaches, personalized vaccines have shown promising results in several clinical trials. These vaccines must be designed computationally in order to cover the enormous variations existing in the diseases and in the patients themselves. Current methods are lacking in one of several aspects, as they only focus on a specific part of the design problem, on a specific type of vaccine, or are unable to guarantee optimality of the solution. In this work, we present a new method to design vaccines that does not suffer from any of these limitations: through a holistic view on the design problem, it can find the best solution for the given design constraints. The flexibility of our method allows us to tune the balance of the different design criteria, perform accurate and reliable comparisons among different solutions, and properly evaluate the trade-offs involved.

**Funding:** ED is supported by the Munich School for Data Science (MuDS, https://mu-ds.de/, Award Number HIDSS-0006 from the Helmholtz Association). BS acknowledges financial supported by the Postdoctoral Fellowship Program of the Helmholtz Zentrum Muenchen (https://www.helmholtz-muenchen.de/fellows/index.html). The funders had no role in study design, data collection and analysis, decision to publish, or preparation of the manuscript.

## Introduction

In recent years, vaccines based on T-cell epitopes, so-called epitope-based vaccines (EV), have become wildly used as therapeutic treatments in case of cancer immunotherapy [1–4] and prophylactically against infectious diseases [5–10]. Compared to regular attenuated vaccines, EVs offer several advantages [11]. Since EVs are based on small peptide sequences, they can be rapidly produced using well-established technologies and easily stored freeze-dried [11]. EVs also do not bare the risk of reversion to virulence as they do not contain any infectious material, and the selection of epitopes can be tailored to address the genetic variability of a pathogen and that of a targeted population or individual increasing its potential efficacy [11].

For an EV to trigger an immune response, the polypeptides have to be delivered to the cytosol of antigen-presenting cells. There, they are digested and cleaved in short pieces by the proteasome or other proteolytic enzymes together with other discarded proteins. Epitopes correctly recovered from the vaccine construct have a chance to bind to the major histocompatibility complex class I (MHC) and be presented on the surface of the cell via the cross-presentation pathway [12]. Naive CD8$^+$ T-cells, recognizing such an epitope-MHC complex, are activated and start circulating, hunting for identical complexes on the surface of infected cells.

To aid the EV design process, bioinformatics approaches have been developed to (1) discover potential candidate epitopes, (2) select a set of epitopes for vaccination, and (3) assemble the selected epitopes into the final vaccine (Fig 1). Three different design principles can be followed to achieve this last step. In epitope cocktails (Fig 1(3a)), the epitopes are not assembled into a polypeptide but delivered separately, effectively eliminating the assembly step altogether. Toussaint *et al.* proposed an approach for epitope cocktail vaccine design that selects a fixed number of epitopes to maximize vaccine immunogenicity using integer linear programming (ILP) [13]. Lundegaard *et al.* proposed a greedy algorithm for epitope selection to maximize antigen and population coverage using a sub-modular function formulation [14].

A second design principle assembles the epitopes in a so-called string-of-beads vaccines (Fig 1(3b)), in which the epitopes are connected directly or by short spacer sequences into long polypeptide sequences. Vider-Shalit *et al.* for example developed a genetic algorithm that selects epitopes to maximize the coverage of viral and human variation while simultaneously optimizing the ordering of the string-of-beads to increase efficacy [15]. Toussaint *et al.* extended their previous framework to find the optimal string-of-beads ordering based on a traveling salesperson problem (TSP) embedding [16], which has been recently extended by Schubert & Kohlbacher to incorporate optimal spacer sequences as well [17].

Recent studies suggest that through the usage of artificial proteins of overlapping epitopes, so-called mosaic vaccines (Fig 1(3c)), both depth and breadth of the T-cell response can be remarkably increased [5–10, 18]. Mosaic vaccines constitute an interesting alternative to string-of-beads EVs, as they incorporate many more epitopes within the same vaccine length

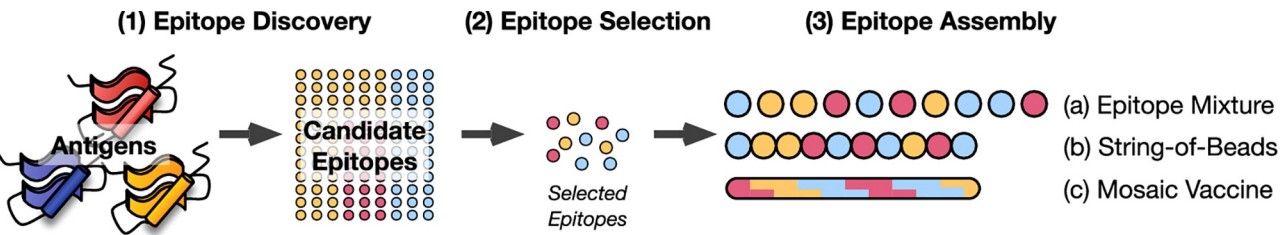

**Fig 1. Epitope-based vaccine pipeline.** An epitope-based vaccine pipeline is comprised of three major steps: (1) epitope discovery, (2) selection, and (3) assembly. Different types of vaccine design exist: (3a) epitope mixture, (3b) string-of-beads, and (3c) mosaics.

[19]. This is especially useful for vaccine development against highly polymorphic viruses like Influenza or the Human Immunodeficiency Virus (HIV). A single mosaic vaccine can be designed to cover the observed variability of the virus by targeting multiple antigens, thereby increasing the potential of obstructing virus escape pathways [7]. To aid the design of such mosaic vaccines, Fischer *et al.* introduced a genetic algorithm that constructs a mosaic protein maximizing the number of nine-mer peptides of an antigen pool [20].

Although multiple algorithms exist to aid EV design, they either lack a theoretical foundation, or only model a sub-problem of the entire design problem. Algorithms in the former category cannot provide any guarantees on the quality of the solution and can be arbitrarily far away from the optimal one, which makes comparisons of different designs potentially unreliable. Algorithms in the latter category, in contrast, are unable to capture the trade-offs between different design stages, thus limiting the space of EV design that can be explored.

In this work, we, therefore, develop a rigorous mathematical formulation that models the entire design process—from epitope selection to assembly—unifies all EV design principles, and can be solved to optimality at the expense of potentially higher computational cost. We then use this framework to explore the trade-off between optimal epitope selection and optimal epitope assembly for string-of-beads vaccines. We also design a cocktail of short polypeptides that have the same properties of a much longer vaccine. We study mosaic vaccines, showing their advantages over string-of-beads in terms of immunogenicity, coverage, and conservation, and give recommendations on which settings to tune for designing effective mosaic EVs. We finally analyze the robustness of our framework in terms of the different required inputs, and show how to reduce the computational burden with little compromise on the quality of the solution.

## Results

### A graph theoretical formalism combines all EV design principles

The first step in a vaccine design pipeline is to obtain a set of potential epitopes to be included in the vaccine. These epitopes are short subsequences derived from antigens of pathogens targeted by the vaccine, identified by experiments or via computational prediction methods. In this work, we focus on MHC class I and use computational models to identify epitopes of length nine, but the framework remains independent from the method of epitope discovery and peptide length.

All three EV design principles try to maximize the vaccine's potential to invoke a strong immune response. However, each design principle does so by focusing on different aspects: epitope mixture vaccines seek a subset $P$ of $k$ epitopes that together have the highest chance of invoking an effective immune response $I(P)$. Similarly, the string-of-beads design problem seeks to find a polypeptide comprised of $k$ concatenated epitopes that simultaneously maximize the vaccine efficacy $I(P)$ and the recovery likelihood of each epitope by the proteasome, which is influenced by the ordering of the epitopes in the construct [21]. In contrast, the mosaic design problem is concerned with constructing an artificial antigen $P$ of fixed length $h$ comprised of potentially overlapping epitopes with maximal efficacy $I(P)$. These design principles can be further generalized to allow the composition of a cocktail of $n$ separate polypeptides $P_1, \ldots, P_n$ that jointly optimize the vaccine efficacy. These three design principles can be unified under a single mathematical framework based on a graph encoding of the optimization problem (Fig 2).

We formulate the generalized EV design problem as a combinatorial optimization problem on a weighted, directed graph $G(V, E, w)$, where the vertices $V$ represent the epitopes and the weight $w(\cdot)$ of the edges $E$ determine the design of the EV. We also add an artificial node $s$

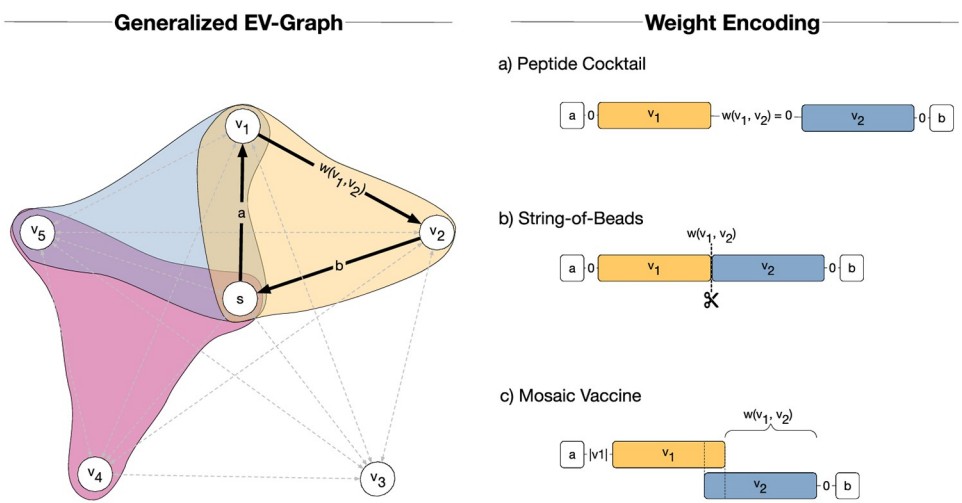

**Fig 2. The graph encoding of the vaccine design problem.** Vertices represent epitopes and edge weights are design-dependent. By jointly modeling the epitope selection and vaccine assembly problem, we seek $n$ subsets of vertices ($n = 3$ in the figure), each representing a separate polypeptide (blue, yellow and red in the figure), with the highest immunogenicity, whose simple tours start from and end in a placeholder node $s$, and are shorter than given limits in terms of the number of vertices $k$ and edge-weight sum $h$. (a) The edge weights are simply ignored for epitope mixtures. (b) For string-of-beads, the edge weights represent the negative log-likelihood of being cleaved at the junction site of the two connecting epitopes, so that low total edge weight is achieved by selecting epitopes that are likely to be separated correctly upon proteasomal cleavage. (c) The weights in mosaic designs represent the added length to the mosaic vaccine once the two connecting epitopes are joined with maximum overlap, so that low total edge weight results in shorter polypeptides.

representing the N- and C-termini of the polypeptides, connecting it to every vertex $v \in V$ such that $w(e_{sv}) = a$ and $w(e_{vs}) = b$, with design-dependent weights $a, b \in \mathbb{R}$. To find the optimal EV in $G(\tilde{V}, \tilde{E}, w)$, with $\tilde{V} = V \cup \{s\}$ and $\tilde{E} = E \cup \{(s, v), (v, s) | v \in V\}$, we seek $n$ subsets $P_1, \ldots, P_n \subseteq \tilde{V}$ of size at most $k + 1$ that only intersect at $s$ and together maximize the vaccine's immunogenicity $I : 2^V \rightarrow \mathbb{R}$. Furthermore, their simple tours $H(P_1), \ldots, H(P_n)$ start and end at $s \in \tilde{V}$ and weigh at most $h \in \mathbb{R}$ (Eq 1). Here, we use the term *simple tour* to refer to a closed walk with no repeated vertices except for $s$, which only appears in first and last position.

$$
\begin{aligned}
\text{Maximize} \quad & I\left(\bigcup_{i=1}^{n} P_i \setminus \{s\}\right) \\
\text{Subject to} \quad & \sum_{e \in H(P_i)} w(e) \leq h && i \in [1, n] \\
& |P_i| \leq k + 1 && i \in [1, n] \\
& P_i \cap P_j = \{s\} && i \neq j \in [1, n] \\
\text{Where} \quad & P_i \subseteq \tilde{V} && i \in [1, n] \\
& I(P) \text{ is the immunogenicity of } P \subseteq V \\
& H(P_i) \text{ is a simple tour visiting the vertices in } P_i && i \in [1, n]
\end{aligned}
\tag{1}
$$

Similar constrained design problems appear in genome assembly [22], and have been extensively studied in the field of operations research under the name of price collecting traveling

salesperson, bank robber, or team orienteering problem [23]. Importantly, they are known to be NP-hard [24].

In the absence of comprehensive vaccine design benchmarks that would allow investigating different contributing properties to vaccine efficacy, we define the vaccine immunogenicity $I$ ($P$) vaguely as the ability to induce a broad immunization in a target population represented by a set of prevalent MHC molecules against a given polymorphic antigen pool. However, the present framework and theoretical results are agnostic to the choice of immunogenicity function.

Therefore, we used a simple linear functional form for $I(P)$ as proposed by Toussaint *et al.* [13], which assumes that each epitope contributes independently to the vaccine's overall immunogenicity with respect to the target population represented by a set of MHC alleles:

$$I(P) := \sum_{v \in P} \sum_{a \in A} p_a i_{va} \tag{2}$$

where $p_a$ is the observed frequency of MHC allele $a \in A$ within the target population and $i_{va}$ is the individual immunogenicity generated by epitope $v$ bound to MHC molecule $a$. We approximate the latter with the log-transformed $IC_{50}$ binding strength between the epitope and the MHC complex, as no sufficiently accurate T-cell reactivity prediction models currently exist. The binding strength can predicted by machine learning algorithms such as NetMHCpan [25], PickPocket [26], or MHCflurry [27].

**Adaptations for epitope mixture design.** We ignore the edge weight constraint by setting $h = \infty$, and the edge weights $w(e_{ij}) = 0$ for $e_{ij} \in \tilde{E}$ (Fig 2a). The size $k$ of $P$, however, has to be defined. Solving the resulting optimization problem yields the epitope mixture with the highest immunogenicity. Note that this is equivalent to the framework proposed by Toussaint *et al.* [13].

**Adaptations for string-of-beads design.** We interpret the edge weights $w(e_{ij})$ as proportional to the negative proteasomal cleavage log-likelihood between epitopes $v_i$ and $v_j$, and set the incoming and outgoing edges of node $s$ to $w(e_{sv}) = w(e_{vs}) = 0$ for every $v \in V$ (Fig 2b), following [16]. The proteasomal cleavage likelihood of two joined epitopes can be predicted with existing proteasomal cleavage site methods such as ProteaSMM [28], PCM [29], or NetChop [30, 31]. The definition of cleavage likelihood can also be adapted to include pre-designed spacers for all possible epitope pairs, for example following Schubert & Kohlbacher's framework [17].

In the following, we refer to the negative sum of edge weights for a string-of-beads EV as "cleavage score", implying that a lower sum is desirable as it corresponds to a larger likelihood of cleavage.

**Adaptations for mosaic design.** We define the edge weight $w(e_{ij})$ as the length that would be added to the mosaic antigen once $v_i$ and $v_j$ are joined at their longest suffix-prefix overlap (Fig 2c):

$$w(e_{ij}) := \begin{cases} |v_j| & \text{if } v_i = s \\ 0 & \text{if } v_j = s \\ |v_j| - \max \{l \in \mathbb{N} | v_i[l:] = v_j[:l]\} & \text{otherwise} \end{cases} \tag{3}$$

where $|v_j|$ represents the length of the epitope sequence $v_j$ and $v_i[l:], v_j[:l]$ represent the first and last $l$ amino acids of epitopes $v_j$ and $v_i$ respectively. Note that Eq 3 can be computed efficiently in time $O(m + k^2)$ for a given set of $k$ strings of total length $m$ by using generalized suffix trees [32].

**Table 1. Integer linear program formulation of the generalized epitope vaccine design problem.** It results in a cocktail of $|T|$ polypeptides each composed of at most $k$ epitopes. Together, the polypeptides cover at least $\Theta_s$ pathogens and $\Theta_a$ MHC alleles, and contain epitopes with an average pathogen conservation of at least $\Gamma$.

| | | | |
|---|---|---|---|
| **Maximize (OBJ)** | $I(P) := \sum_{t \in T} \sum_{v \in V} \sum_{a \in A} y_{vt} p_a i_{va}$ | | Overall immunogenicity |
| Subject to | | | |
| (C1) | $\sum_{v \in \tilde{V}: v \neq w} x_{wvt} = \sum_{v \in \tilde{V}: v \neq w} x_{vwt} = y_{wt}$ | $\forall w \in \tilde{V}, t \in T$ | Consistency between $x$ and $y$ |
| (C2a) | $u_{vt} - u_{wt} + 1 \leq (|V| - 1)(1 - x_{vwt})$ | $\forall e_{vw} \in E, t \in T$ | Subtour elimination |
| (C2b) | $1 \leq u_{vt} \leq |V| - 1$ | $\forall v \in V, t \in T$ | Bounds for node potential |
| (C3) | $\sum_{t \in T} y_{vt} \leq 1$ | $\forall v \in V$ | Tours can only share $s$ |
| (C4) | $\sum_{t \in T} \sum_{v \in V} x_{svt} = \sum_{t \in T} \sum_{v \in V} x_{vst} = |T|$ | | All tours leave from and return to $s$ |
| (C5) | $\sum_{(v,w) \in E} x_{vwt} w(e_{vw}) \leq h$ | $\forall t \in T$ | Max. tour edge weight is $h$ |
| (C6) | $\sum_{v \in V} y_{vt} \leq k$ | $\forall t \in T$ | Max. tour vertex count is $k$ |
| (C7a) | $\sum_{t \in T} \sum_{v \in V} y_{vt} \tau_{vs}^{(S)} \geq \theta_s$ | $\forall s \in S$ | Is pathogen $s$ covered? |
| (C7b) | $\sum_{s \in S} \theta_s \geq \Theta_s$ | | Cover at least $\Theta_s$ pathogens |
| (C8a) | $\sum_{t \in T} \sum_{v \in V} y_{vt} \tau_{va}^{(A)} \geq \theta_a$ | $\forall a \in A$ | Is allele $a$ covered? |
| (C8b) | $\sum_{a \in A} \theta_a \geq \Theta_a$ | | Cover at least $\Theta_a$ alleles |
| (C9) | $\sum_{t \in T} \sum_{v \in V} y_{vt} \left( \sum_{s \in S} \tau_{vs}^{(S)} - \Gamma \right) \geq 0$ | | Avg. epitope conservation at least $\Gamma$ |
| (C10a) | $x_{vwt}, y_{vt}, \theta_a, \theta_p \in \{0, 1\}$ | $\forall v, w, t, o$ | Variable domains |
| (C10b) | $u_{vt} \in \mathbb{N}$ | $\forall v \in V, t \in T$ | |
| Where: | | | |
| $V, T, A, S$ | Indices of epitopes, tours, alleles, and pathogen sequences | | |
| $x_{vwt}, y_{vt}$ | Binary decision variables for edges and epitopes | | |
| $p_a$ | Observed frequency of allele $a$ in the population | | |
| $i_{va}$ | Log-transformed IC$_{50}$ binding strength between epitope $v$ and allele $a$ | | |
| $h$ | Maximum edge weight for each tour | | |
| $k$ | Maximum vertex count for each tour excluding $s$ | | |
| $\tau_{va}^{(A)}, \tau_{vs}^{(S)}$ | Indicator variables, equal 1 iff epitope $v$ covers allele $a$/pathogen $s$ respectively. | | |
| $\Theta_a, \Theta_p$ | Minimum number of alleles/pathogens to cover or zero if not relevant. | | |
| $\Gamma$ | Minimum average conservation or zero if not relevant. | | |

The edge-weight sum of any tour that starts and ends at $s$ is then equal to the length in amino acids of the resulting mosaic sequence, so that the number of epitopes in the vaccine can be increased by finding pairs of epitopes with a high degree of overlap. The solution to the so defined problem is a polypeptide comprised of overlapping epitopes with optimal immunogenicity whose length is at most $h$.

## Formulation as an integer linear program guarantees optimality

With the aforementioned definitions, we can formulate the generalized EV design problem as an integer linear program (ILP) encoding the team orienteering problem [23] (Table 1). This guarantees to construct an optimal EV at the cost of potentially long run times and/or memory requirements, since the number of variables and constraints grows quadratically with the

number of epitopes in consideration, usually in the order of $10^3$ or $10^4$. Note that, if desired, other optimization methods can be employed to solve the generalized EV design problem we formulated in Eq 1, however potentially losing optimality guarantees.

Every tour represents a distinct polypeptide in the vaccine. We introduce binary decision variables $x_{vwt}$ and $y_{vt}$ indicating whether the tour $t$ visits the edge between $v$ and $w$ and the vertex $v$, respectively. We enforce the consistency between these decision variables with the constraint C1, which also ensures that for every vertex the number of chosen incoming edges equals the number of chosen outgoing edges. The constraints C2a and C2b exclude the case of a tour $t$ containing a set of edges forming disjoint subtours. Together with C1, this forces each tour to be connected and visit $s$. We are using the Miller-Tucker-Zemlin (MTZ) formulation [33] for C2, but other options with different computational properties exist. MTZ associates a node potential $u_{vt} \in \mathbb{N}$ to each vertex in $V$ (i.e., not including $s$) and constrains edges in a tour to connect nodes with increasing potential. This forces every tour to pass from $s$ to "reset" the potential, allowing it to decrease from the last vertex before $s$ to the first after $s$. We also have to ensure that every node is visited at most once across all tours (C3) and that each tour starts from and ends in $s$ (C4). C2 makes C4 redundant, but alternative subtour elimination constraints may still need it. We then constrain the edge weight limit (C5) and the length (C6) of each tour. Constraints of the form C7 and C8 can be used to enforce a minimum overall coverage of $\Theta_s$ different pathogens and $\Theta_a$ MHC alleles among all tours, whereas C9 can be used to enforce a minimum average conservation (number of antigens covered by each epitope) of $\Gamma$. C7 and C9 require a set $S$ with the available pathogen sequences and indicator variables $\tau_{vs}^{(S)}$ specifying whether epitope $v$ covers pathogen $s$. Similarly, C8 requires the set $A$ of MHC alleles and indicators $\tau_{va}^{(A)}$.

## Jointly approaching epitope selection and assembly captures the trade-off between cleavage likelihood and immunogenicity

Vaccines are cleaved by the proteasome, and the resulting peptides are eventually presented on the cell surface by the MHC class I complex. A string-of-beads vaccine is effective only if the proteasome correctly cleaves the epitopes contained in the vaccine. Wrong cleavage sites would result in new, unwanted peptides with unknown properties, thereby decreasing the efficacy of the vaccine. This risk can be managed with our framework by appropriately setting $h$, the maximum total negative cleavage log-likelihood between all epitopes of the vaccine. Interpreting and predetermining this quantity is, however, difficult. We might not even be interested in a solution with a fixed $h$, but rather want to explore the inter-dependencies between the immunogenicity objective and the cleavage likelihood of the string-of-beads EV. This leads to a reinterpretation of the design formulation as bi-objective optimization problem, in which we simultaneously optimize the overall immunogenicity $I(P)$ and the length of the tours $H(P_i)$, $i = 1, \ldots, n$ (i.e., the overall cleavage likelihood). We then explore the Pareto frontier of this problem with the augmented $\epsilon$-constraint method [34] (Section A in S1 Appendix).

For this and all later experiments, predicted epitopes for the 27 most prevalent MHC class I alleles (from Toussaint *et al.* [16]) were extracted from 1,917 sequences of the HIV-1 Clade B and C Nef proteins, for a total of 52,712 epitopes. We then created five bootstraps of 300 randomly chosen sequences (see Materials and methods for details).

For each bootstrap, we selected the 1,000 epitopes with the largest immunogenicity and computed the edge weights that resulted from directly joining the epitopes and from joining the epitopes with the optimal spacers using Schubert & Kohlbacher's framework [17] (see Materials and methods for details). Because the edge weights computed by these two methods are not directly comparable, we separately normalized the cleavage scores by the maximum

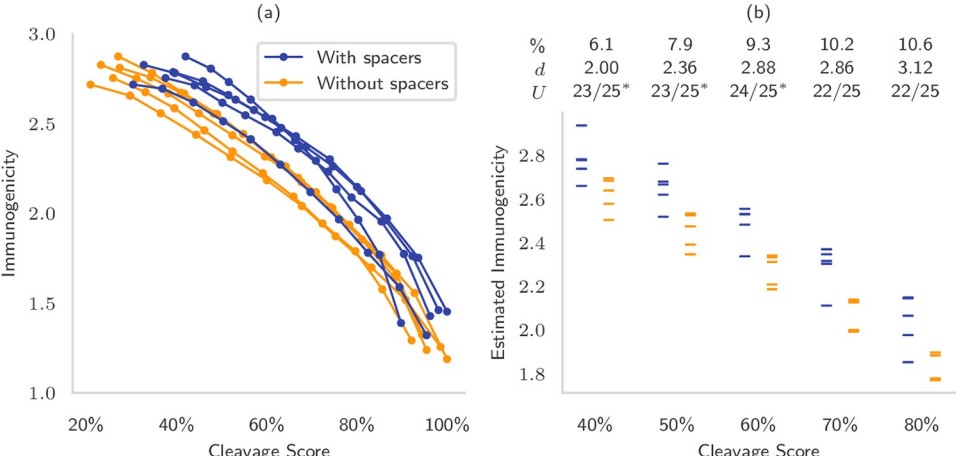

**Fig 3. Novel EV design possibilities obtainable by considering epitope selection and epitope assembly at the same time.** (a) We created five bootstraps of protein sequences and generated string-of-beads vaccines on the Pareto frontier by maximizing immunogenicity and cleavage score at the same time. The epitopes were either joined directly (yellow) or by optimal spacer sequences (blue). The cleavage score of the two groups was normalized separately to allow comparisons between spacers and direct links. (b) We selected five percentiles of the cleavage score, from 40 to 80 of the global maximum, and estimated the immunogenicity that can be obtained via linear interpolation between the closest points on the Pareto frontier of each bootstrap. The table above shows the percent increase in immunogenicity relative to the no-spacers group, the effect size $d$ computed as the difference of the means normalized by the standard deviation of the no-spacers group, and the Mann-Whitney $U$ test statistic, that shows the number of pairwise comparisons that were favorable for the designed spacers (* for $p < 0.05$).

score that was achieved by each method, and compared percentiles instead. We then generated eleven Pareto-efficient solutions, each of them ten epitopes long, to illustrate the trade-off between optimizing cleavage and immunogenicity (Fig 3a).

Without spacers, there was an almost linear relationship between the decrease in cleavage score and immunogenicity, so that even modest improvements in cleavage (immunogenicity) required significant sacrifices in immunogenicity (cleavage). In particular, 80% of the maximum cleavage score could only be achieved with 65% of the maximum immunogenicity on average, and 80% of the maximum immunogenicity with 59% of the maximum cleavage score on average. These numbers improved with optimal spacers so that cleavage score at 80% still achieved 71% of the maximal immunogenicity and immunogenicity at 80% reached 73% maximal cleavage score. Using spacers provided an increase in immunogenicity between 6% and 11% for the same percentile cleavage score, corresponding to an effect size between two and three (Fig 3b). This again highlights the need for spacers, either fixed or suitably designed, joining epitopes to further increase the cleavage likelihood.

Sequential approaches such as OptiTope [13] are only able to generate the highest-immunogenicity solutions, as they do not consider the subsequent assembly phase at all. These methods simply cannot be used to balance the quality of selection and assembly except by direct enumeration of all possible sequences of $k$ epitopes, which in this case would be $\prod_{i=0}^{k}(N - k) \approx 3 \cdot 10^{45}$ with $k = 10$ and $N = 13,500$, the approximate number of epitopes in our bootstraps.

To illustrate how ordering affects the cleavage likelihood and how it is optimized by our framework, we randomly shuffled the epitopes of the vaccines without spacers on the Pareto frontier of each bootstrap and compared the cleavage scores of the resulting string-of-beads construct with the score of the optimized vaccine (Fig 4).

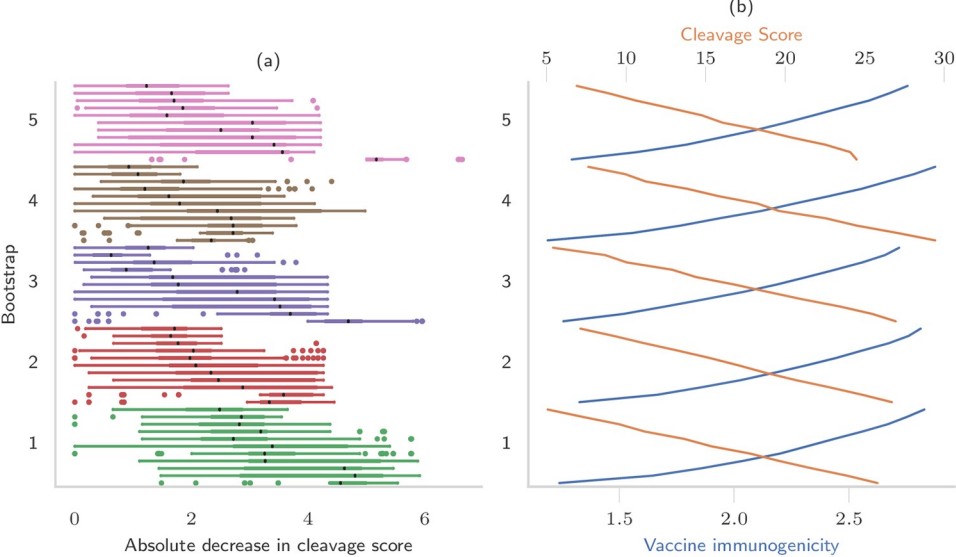

**Fig 4. Decrease in cleavage score by randomly permuting string-of-beads vaccines.** (a) For each vaccine in the Pareto frontier of each bootstrap, we created 50 new string-of-beads by randomly permuting its epitopes and compared the cleavage scores of the two vaccines. The black dots in the box plots show the median decrease. (b) cleavage score and immunogenicity of the original vaccine with optimized ordering.

The overall decrease was between 0 and 6.65 with median 2.32 (50% interval: [1.39, 3.51]), corresponding to a percent decrease between 0 and 49.6% with median 13.1% (50% interval: [8.33-17.83]). The median reduction in cleavage score was correlated with the optimized cleavage score (Spearman's $\rho = 0.74$, $p = 5 \cdot 10^{-11}$), so that the largest decrease was observed when shuffling epitopes of vaccines designed with greater emphasis on cleavage score. There was no correlation between the ordering of the epitopes and the score difference ($\rho = 0.01$, $p = 0.55$), where the ordering was quantified by the number of identical pairs of adjacent epitopes in the optimized and shuffled vaccines. However, the difference was negatively correlated with the number of epitopes that did not change position when shuffling ($\rho = -0.24$, $p = 7 \cdot 10^{-38}$). Following our definition of edge weight, we can consider a negative weight as indicating a cleavage site (see Materials and methods section). Among the vaccines on the Pareto frontier, 96.3% had nine cleavage sites, the maximum possible, on the epitope junctions, and the remaining had eight. Among the shuffled vaccines, only 50.9% had nine junction cleavage sites, 41.1% had eight and the remaining 8.0% had six or seven.

No shuffled vaccine had a better cleavage score than the original vaccine on the Pareto frontier, although some had comparable scores. This follows from the definition of Pareto frontier: each point can be obtained by maximizing the edge weight subject to a certain bound on the immunogenicity. This implies that the vaccines on the Pareto frontier have the largest edge weight among all possible orderings of the epitopes they contain. Using different epitopes can increase the immunogenicity or the cleavage score, but not both.

## Joint design of polypeptide cocktails increases immunogenicity without sacrifices

A large number of epitopes might be necessary to create a vaccine achieving extremely high conservation and/or coverage. Long polypeptides, however, are harder to manufacture [35, 36] and, in practice, most synthetic vaccines tested so far are composed of sequences of 10 to 50

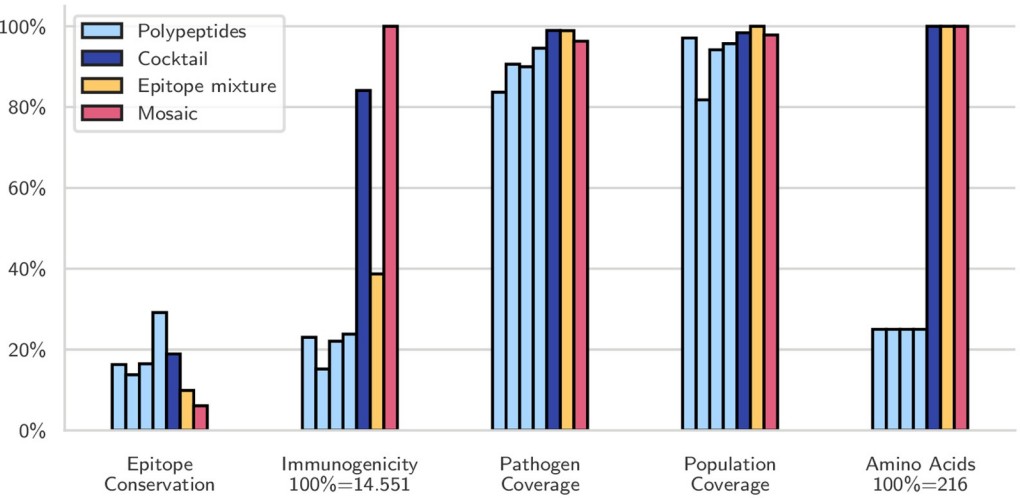

**Fig 5. Cocktail of mosaic proteins compared to an equivalent string-of-beads vaccine.** We designed a cocktail (blue) of four polypeptides (cyan) that covers 99% of the pathogens, even though the single fragments only cover between 85 and 95%. The orange and red bars correspond to an epitope mixture designed by OptiTope and a mosaic with the same number of amino acids respectively; the former was required to reach 99% pathogen coverage, and the latter was unconstrained.

amino acids [3, 37–40]. Our framework can be used to design a vaccine that meets such extreme requirements on coverage and/or conservation by creating several short polypeptides that are optimized simultaneously and can be synthesized in parallel for a fraction of the cost and time.

For example, an epitope mixture designed by OptiTope [13] needs at least 24 epitopes to cover 99% of the complete set of pathogens. Concatenating these 24 epitopes in a single string-of-beads construct results in a 216 amino acid-long polypeptide, which may be undesirably lengthy. We, therefore, designed a vaccine composed of four separate mosaic polypeptides of 54 residues each, so that the total number of amino acids remained unchanged. We also designed a single mosaic of 216 amino acids with no coverage enforced as a baseline. To keep the computational requirements at acceptable levels, the mosaic cocktail was designed on epitopes coming from the complete set of 1,917 sequences, but restricted to the union of the 2,000 epitopes with the highest immunogenicity and the 2,000 with the highest pathogen coverage.

The resulting four polypeptides together had roughly twice the conservation and the immunogenicity as the epitope mixture (Fig 5). Most notably, none of them reached the required pathogen coverage in isolation; only when considered jointly they covered the required number of pathogens. The unconstrained single mosaic vaccine already covered 96% of the pathogens even though its epitopes had the lowest conservation among the three vaccines. Unsurprisingly, it also had the largest immunogenicity, almost 20% more than the polypeptide cocktail and 260% more than the epitope mixture, simply because it included so many more epitopes (208, compared to 184 for the cocktail and 24 for the mixture).

## Mosaic design greatly increases immunogenicity and pathogen coverage compared to string-of-beads

We compared epitope mixtures to mosaics of the same length with respect to immunogenicity, conservation, pathogen and population coverage as we increased the number of admissible amino acids in the vaccine. The metrics for an epitope mixture are upper bounds for what can be achieved by a string-of-beads EV, as the cleavage requirements would only reduce the set of

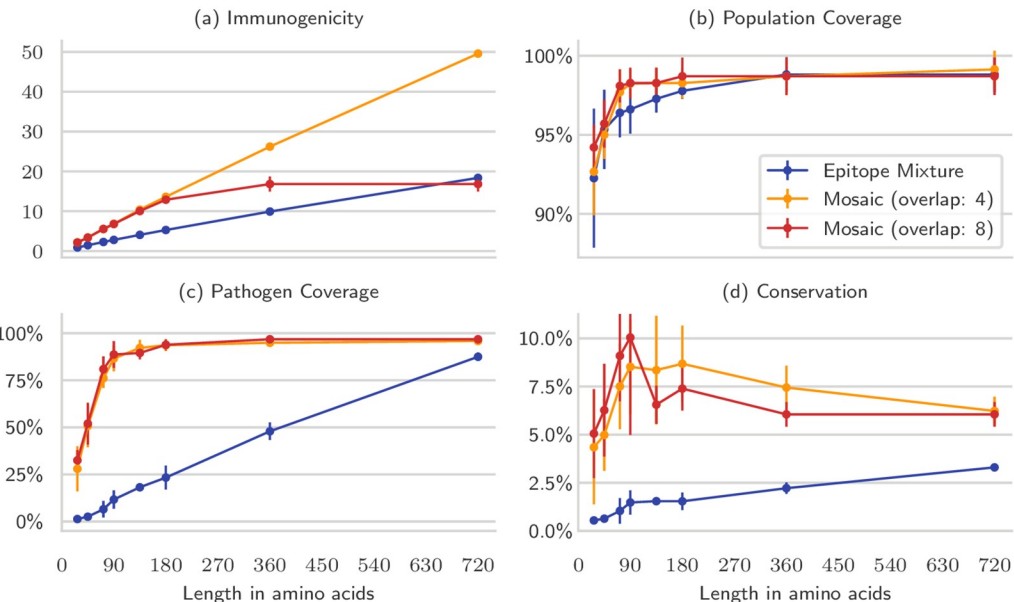

**Fig 6. Comparison of mosaic and epitope mixture vaccines of different sizes.** Mosaic vaccines were much better than epitope mixtures or string-of-beads of the same length (blue), as long as the pathogens offer enough epitope variety. By enforcing an overlap between epitopes of eight amino acids (red), the vaccine did not improve after a certain length. This could be prevented by relaxing this requirement to only four amino acids (yellow). The vaccines are compared with respect to four metrics: immunogenicity (a), population coverage (b), pathogen coverage (c), and conservation (d). The experiment was repeated for each of the five bootstraps, and bars represent the resulting standard deviation. Note that these vaccines were not designed with pathogen coverage nor epitope conservation in mind; mosaics naturally reached higher values.

eligible epitopes that can be included in the vaccine. The mosaics were designed by enforcing minimum overlaps of four and eight amino acids, and the experiment was repeated for each bootstrap.

By leveraging overlaps, the mosaic designs were able to include more epitopes in the same number of amino acids, resulting in improved immunogenicity, pathogen coverage and conservation compared to the epitope mixtures (Fig 6). The mosaic with overlap of eight amino acids, however, could not be improved substantially after 172 epitopes (180 amino acids), and plateaued by 360 amino acids. This happened because there were not enough pairs of suitably overlapping epitopes to produce longer polypeptides. Relaxing the overlap requirements to at least four amino acids allowed the framework to produce pseudo-mosaic vaccines that contained less epitopes than the theoretical maximum, but had, nonetheless, much higher immunogenicity than the epitope mixtures. It is also evident that mosaic vaccines could inherently reach higher pathogen coverage with shorter polypeptides, even though no such constraint was imposed on the design. Conservation, however, remained mediocre.

## Short mosaic vaccines achieve very high coverage

We designed mosaic vaccines composed of a single polypeptide on each of the five bootstraps following the genetic algorithm introduced by Fisher *et al.* [20, 41], using the recommended parameter settings (notably, unlike our framework, the length of the polypeptides cannot be specified). We then used our framework to design a mosaic vaccine of the same length (206 amino acids), with at least the same pathogen and population coverage and epitope conservation. This resulted in very similar mosaics with essentially the same properties, covering the

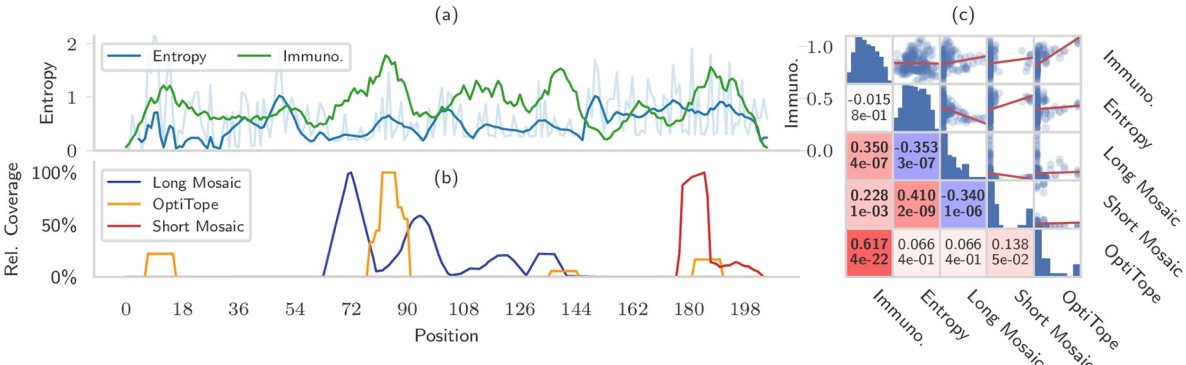

**Fig 7. Mosaic vaccines naturally target conserved regions even when this is not required.** (a) shows, for each residue position in aligned sequences where the consensus is not a gap, the smoothed entropy (blue, and residue entropy in lighter color) and the potential immunogenicity (green) (b) shows the number of pathogens covered in each position by a 20-epitopes mixture with maximal immunogenicity (yellow), a short mosaic of 28 amino acids (red) and a long mosaic of 90 amino acids (blue). The count is normalized separately for each vaccine to account for their different coverage. (c) shows the pairwise correlations of the variables shown in the left plot, so that every dot in the scatter plots corresponds to a different residue position, and linear fits are shown in red. The lower triangular half shows the Spearman correlation coefficients (above) and the respective *p*-value (below). Colors range from blue (large negative correlation) to white (no correlation) to red (large positive correlation), and the font is bold if the correlation is significant with a confidence of at least 99.5% at the Bonferroni-corrected significance level of 5%. The diagonal contains histograms showing the distribution of each variable, with logarithmic *y* axis.

same 26 MHC alleles and 99.1% of pathogens, an average epitope conservation of 28%, and immunogenicity of 10.8. However, Fisher *et al.* optimize for coverage, not for immunogenicity. When we did the same, we were able to achieve the same properties with a reduction in epitopes of about 50%, indicating the Fisher *et al.*'s solution was sub-optimal (the alternative ILP formulation is in Table A of S2 Appendix).

## Long mosaic vaccines inherently target conserved regions

The previous experiments showed that mosaic vaccines can reach very good pathogen coverage with ease. Intrigued by this characteristic, we studied and compared the exact positions covered by the epitopes of mosaic and string-of-beads vaccines.

The vaccines were designed on the complete pathogen set. The string-of-beads contained 20 epitopes, the short mosaic vaccine 28 amino acids, and the long mosaic vaccine 90 amino acids. We then aligned the pathogen sequences using MAFFT [42] and counted how many epitopes in the EVs covered each position. We also computed the potential immunogenicity of a position as the sum of the immunogenicities of all the epitopes covering that position, and used position-specific entropy to quantify the variation among sequences [43] (Section B in S1 Appendix). Finally, we ignored the positions where the consensus sequence was a gap.

Analyzing the epitopes included in the vaccines showed that they did not appear in random positions of the pathogens, but were concentrated in a few distinct regions that differed between vaccines (Fig 7b). It was evident that string-of-beads vaccines, cleavage requirements aside, targeted the most immunogenic regions with no regards for their conservation, whereas mosaic vaccines, especially longer ones, preferred to focus on conserved regions. These correlations, as quantified by the Spearman coefficient ($\rho$), were generally weak or moderate, but statistically significant (Fig 7c). The epitope mixture's coverage was well correlated with immunogenicity ($\rho = 0.617$, $p = 4 \cdot 10^{-22}$), but not with entropy ($\rho = 0.066$, $p = 4 \cdot 10^{-1}$). The long mosaic vaccine sought immunogenic ($r = 0.350$, $p = 4 \cdot 10^{-7}$), but low entropy regions ($\rho = -0.353$, $p = 3 \cdot 10^{-7}$). Interestingly, the short mosaic vaccine covered an entirely different region and was correlated with both immunogenicity ($\rho = 0.228$, $p = 1 \cdot 10^{-3}$) and entropy

($\rho = 0.410$, $2 \cdot 10^{-9}$). Entropy and immunogenicity are, curiously, not correlated ($\rho = -0.015$, $p = 8 \cdot 10^{-1}$).

## Mosaic vaccines should be designed with epitope conservation in mind

There is growing evidence that effective epitope vaccines for highly variable viruses such as HIV, Hepatitis C Virus (HCV), as well as diseases such as Malaria, Cancer, and Influenza should target conserved epitopes [44–48]. Figs 6 and 7 clearly show that mosaic vaccines have a natural tendency to spontaneously achieve high pathogen coverage, by targeting conserved regions of the pathogen. We wanted to see how much further we could exploit this behavior by optimizing conservation and coverage.

We modified the ILP formulation to maximize average epitope conservation and pathogen coverage together with immunogenicity (S2 Appendix). Since immunogenicity is a couple of orders of magnitude smaller than the other two, it will only be optimized when further improvements in conservation or coverage are practically insignificant. We then compared mosaic vaccines of growing sizes optimized against these three criteria.

The average epitope conservation could be greatly improved until around 40%, and it came with increased pathogen coverage compared to mosaic vaccines optimized for immunogenicity (Fig 8). Most epitopes had very poor conservation, which means that optimizing its average becomes harder as the vaccine size increases. In fact, longer vaccines had smaller conservation. Moreover, immunogenicity grew more slowly when conservation was optimized: it was on par with pathogen coverage-optimized vaccines for short mosaic designs, but was almost 30% smaller for long mosaics.

This suggests that the best results are obtained with short mosaic vaccines designed to have high average epitope conservation. Besides having considerably larger conservation, both their immunogenicity and their pathogen coverage were still close to the theoretical maximum that can be achieved by explicitly optimizing for them. As the mosaic designs became longer, this gap widened, and so-designed vaccines lost their advantages. However, we have shown previously that long vaccines can be replaced by cocktails of short polypeptides with essentially the same joint properties.

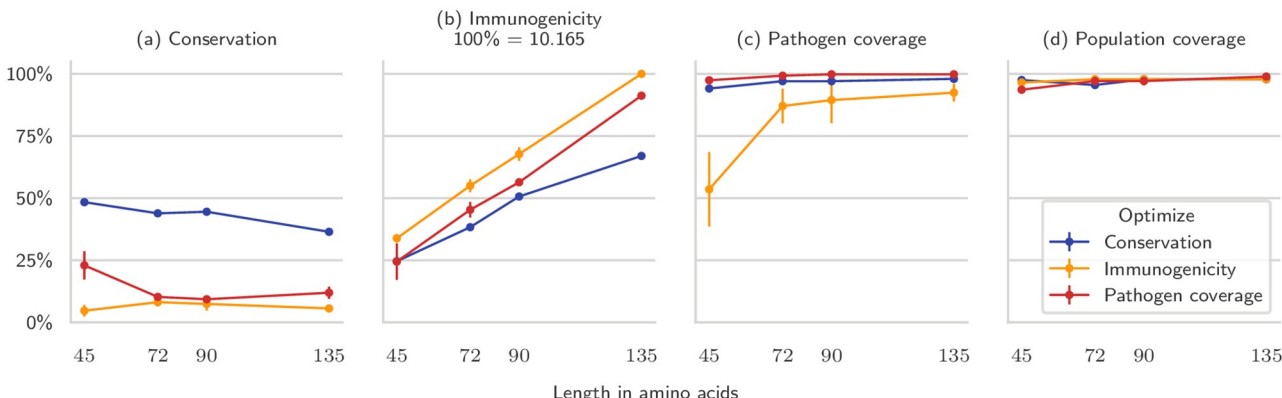

**Fig 8. Comparison of mosaic vaccines optimized for different objectives.** Here we designed mosaics of varying amino acid length (on the *x* axis) while optimizing for conservation (blue), immunogenicity (red), and pathogen coverage (yellow). The plots compare the vaccines in terms of conservation (a), immunogenicity (b), pathogen coverage (c), and population coverage (d). For longer vaccines, optimizing for pathogen coverage only gives modest improvements on the mosaics optimized for immunogenicity in terms of coverage, and does not increase conservation by much. When optimized for, conservation is considerably higher but becomes harder to improve as the vaccine becomes longer, due to the fact that few epitopes are well conserved and highly immunogenic at the same time. Average of five runs, standard deviation on the error bars.

## The framework is robust with respect to different epitope immunogenicity functions

The immunogenicity of the vaccine in Eq 2 is defined as a weighted sum of the immunogenicities of the individual epitopes, appoximated by the $IC_{50}$ binding strength. Several methods exist to predict this quantity, the most well-known being NetMHCpan [25], PickPocket [26], and MHCflurry [27]. Furthermore, the authors of NetMHCpan recommend using the percentile rank as a more accurate indicator of binding, where the rank is computed by comparing the $IC_{50}$ binding strength of an epitope with the binding strengths of random natural peptides.

To test the robustness of the framework for other choices of MHC binding affinity prediction tools, we repeated the Pareto experiment (without spacers) using the four immunogenicity predictors mentioned above and compared the vaccine immunogenicities as computed by the different tools (Fig 9). Since we cast the optimization problem as maximization, we transformed the rank so that 100 represents the largest immunogenicity.

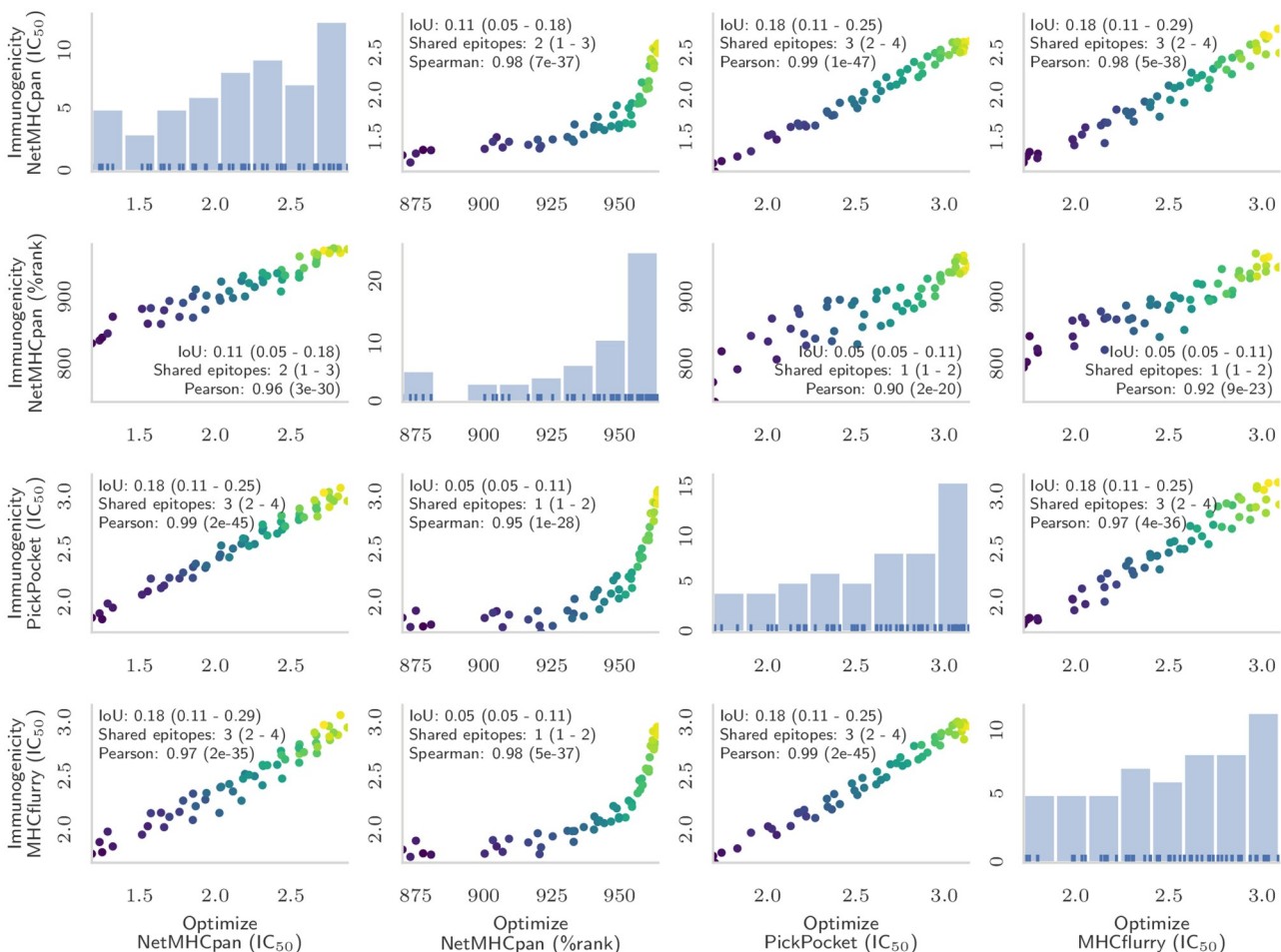

**Fig 9. Effect of different immunogenicity predictors on optimized vaccines.** Each scatter plot shows the immunogenicity predicted by a certain method (*y* axis), when the ten-epitopes string-of-beads vaccine was designed optimizing the immunogenicity predicted by a different method (*x* axis). The diagonal shows the immunogenicity distribution of the optimized vaccines. The color of each point indicates the cleavage score of the vaccine (brighter is larger). Inside each scatter plot, we report the intersection-over-union (IoU) of the epitopes, the number of shared epitopes between vaccines of similar cleavage score indicating median and 25th and 75th percentile in parentheses, and the correlation coefficient (Pearson or Spearman depending on the plot) with its *p*-value against the null hypothesis of no correlation.

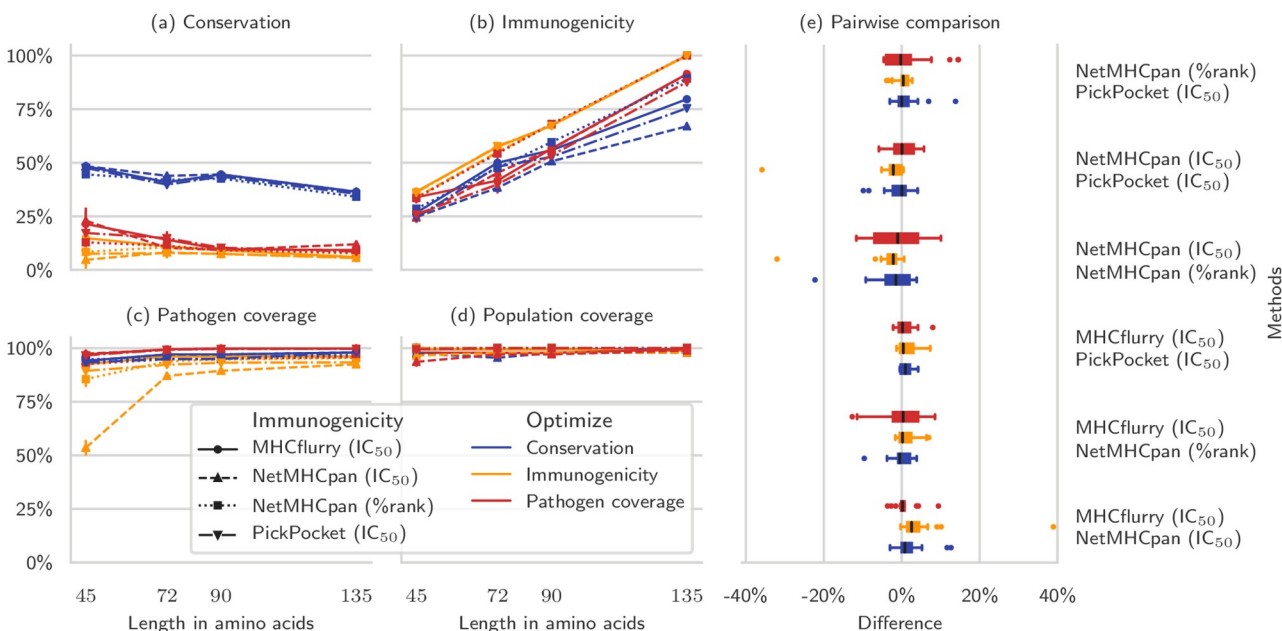

**Fig 10. Effect of different immunogenicity predictors on vaccines optimized for different objectives.** We repeated the experiment shown in Fig 8 using different epitope immunogenicity predictors. For each bootstrap, we designed mosaic vaccines optimizing their conservation (blue), immunogenicity (yellow), or pathogen coverage (red) and compared the vaccines in terms of conservation (a), immunogenicity (b), pathogen coverage (c), and population coverage (d). (e) shows the pairwise difference of the four metrics across all vaccine sizes, separated by optimization objective, between all pairs of immunogenicity functions. There is little variation among the results obtained with different immunogenicities.

Optimizing an $IC_{50}$-based immunogenicity resulted in a very well correlated (0.92 or larger and statistically highly significant, with *p*-values smaller than $2 \cdot 10^{-20}$) and linear improvement in all other immunogenicities. Optimizing the rank-based immunogenicity showed instead a distinct two-phases increase, where at first the $IC_{50}$ immunogenicity barely increased, then quickly gained the lost ground as the rank immunogenicity showed only marginal improvements. Even though the immunogenicities were well correlated, vaccines with similar cleavage scores were composed of mostly different epitopes, sharing only one or two on average and with an intersection-over-union metric between 5% and 18%. This can be explained by the large number of epitopes that had similar immunogenicity. The average number of epitopes that had an immunogenicity score within 0.5% of any given epitope was 43, 46, 62, and 113 for MHCflurry, NetMHCpan ($IC_{50}$), PickPocket and NetMHCpan (rank) respectively. This means that there were in the order of $43^{10}$ sets of ten epitopes that had an immunogenicity score within 5% of a given set of ten epitopes, as quantified by MHCflurry.

We also repeated the experiment in Fig 8 for each immunogenicity predictor, and compared vaccines that were optimized for different metrics across predictors (Fig 10). The average difference between metrics of different predictors was 0.1% with standard deviation of 5.5%, and 50% (90%) of the differences were between -1.7% and 2.1% (-7.9% and 7.2%). Only three outliers had an absolute difference larger than 25%, corresponding to the much lower pathogen coverage achieved at 45 amino acids when optimizing NetMHCpan's $IC_{50}$ immunogenicity (Fig 10c).

## Vaccines designed on small subsets generalize to the full dataset

All previous experiments were obtained by considering 300 random proteins out of a few thousand, except the cocktail in Fig 5, designed on the 4,000 most immunogenic peptides of all

sequences, and the association between positions covered by mosaics and entropy in Fig 7. One naturally wonders whether the vaccines produced on such small subsets were still as good on the general pathogen population.

The only quantities that can change, for a given vaccine, are epitope conservation and pathogen coverage, while designing a vaccine *de novo* can result in higher immunogenicity. We designed a mosaic vaccine of 206 amino acids for each bootstrap, ensuring it covered at least 99.1% of the pathogens, 26 alleles out of 27, and having an average epitope conservation of 28% or more. We then evaluated coverage and conservation relative to the set of all pathogens, and quantified the differences in the evaluation metrics. This revealed that there was no difference on conservation (paired t-test, $t = 0.04$, $p = 0.487$) between the small subsets and the full set, but pathogen coverage slightly decreased ($t = 2.77$, $p = 0.025$, from 99.1%, std. 0.4% to 98.7%, std. 0.2%). A string-of-beads with 10 epitopes and no constraints achieved an immunogenicity score of 3.08 on the full set, just 10% higher than what could be achieved with the same settings on the bootstraps in the Pareto frontier experiment. A 216-amino acids mosaic in the same setting as the cocktail experiment, but designed on all the epitopes, improved immunogenicity by 18% (from 14.55 to 17.23), but worsened coverage by 1% (from 96.3% to 95.0%) and conservation by 41% (from 6.1% to 3.6%), suggesting that subsetting epitopes should be done with greater care.

It might seem surprising that vaccines developed on roughly 15% of the pathogens generalize to a larger population. However, epitopes with high coverage and conservation on a large population are likely to be so also on random subsets of it. In fact, even though each of the five random subsets contains only about 25% of the epitopes found in the full pathogen set, the pairwise overlap was between 46 and 48%, and 27% of the epitopes were shared among all five sets.

## Discussion and conclusion

Epitope-based vaccine (EV) design has thrived in recent years, and multiple design principles have emerged aided by the heavy use of bioinformatics approaches. However, most proposed design algorithms are lacking in one of several dimensions: they model only individual stages of the entire design problem (e.g., [13, 17]), use *ad hoc* heuristics (e.g., [14]), or optimization algorithms that cannot guarantee convergence to the optimal solution (e.g., [15, 20]).

Here, we proposed a graph-theoretical formalism for EV design that models the complete design process and includes every prevalent design principle as special case. We showed how to formulate this optimization problem as an integer linear program to obtain a guaranteed optimal solution. This, in turn, enables informed choices throughout the design process by accurately and reliably determining the trade-offs involved: for example, we precisely quantified the decrease in immunogenicity that has to be paid to achieve gains in other metrics such as coverage and conservation, and showed the advantage of mosaic over string-of-beads designs under our modeling assumptions. In practice, we might be overestimating mosaics' immunogenicity, as our framework does not model their cleavage by the proteasome, which means that we have no control over which epitopes will actually be recovered. However, their successes in recent clinical trials [5–10, 18] suggest that their advantage over string-of-beads is, nonetheless, real.

The framework is general enough to be broadly applicable to several types of vaccines. By tailoring the definition of immunogenicity, both T and/or B cell epitopes can be included in the vaccine, eventually producing cytotoxic T cells and plasma cells respectively as required. Personalized cancer vaccines can be produced by reinterpreting the components of the framework, but no change would be necessary. The input epitopes would be extracted from the

patient's mutanome, which would then act as the pathogens. Pathogen coverage constraints would then be interpreted as coverage of mutated proteins or somatic mutations. The input MHC alleles would match the genotype of the patient. The conservation of epitopes with respect to the mutations would not need to be considered, but conservation with respect to MHC alleles would contribute to the robustness of the vaccine.

Jointly approaching the selection and assembly problems enables the exploration of new possibilities in the EV design space. We demonstrated this by investigating the trade-off between immunogenicity and cleavage likelihood in string-of-beads designs, and by creating the optimal cocktail of mosaic polypeptides for a given coverage whose design would be impossible with iterative, stage-wise optimization methods. Our results also show that it is easy to reach very good population coverage by virtue of our definition of immunogenicity based on MHC binding. Finally, convergence and optimality guarantees of linear programming solvers allow us to find solutions that are, sometimes, substantially better than those found by optimization algorithms that lack these guarantees.

The price to pay for the increased modeling power is increased computational resources needed to solve the graph optimization problem. Being based on the team orienteering problem, EV design with our framework is a NP-hard problem, and the size of the graph grows quadratically with the number of epitopes in consideration. However, we conducted several experiments on subsets of the pathogens or the epitopes and showed that results obtained in this way are only slightly worse than what can be obtained by considering the complete set of pathogens/epitopes. We argued that this is possible because highly conserved epitopes are likely to abound in smaller subsets of the pathogen sequences too. This means that such graphs can easily be pruned, resulting in much smaller problems that can be solved in reasonable time without compromising the quality of the final solution.

Alternatively, the solver can be interrupted early when the current solution is within a few percent of the optimal one. ILP solvers iteratively improve a candidate solution and an upper bound on the objective of the optimal solution at the same time [49]. This gap reflects the maximum distance between the candidate solution and the optimal one: when they match, the current solution is optimal. Empirically, in our setting, most of the time is spent on improving solutions that are only at most 2%-5% away from the optimal one. Therefore, the solver can be safely interrupted early on a good quality solution.

ILP solvers are complex machinery governed by several parameters that affect how they search for a solution, and, therefore, how much time they need. Different types of problems benefit from different parameter settings. Therefore, further gains can be achieved when dealing with a large number of instances by tuning these parameters to reduce the time needed to find a solution [50–52].

Another limitation is that our formulation as an ILP limits the expressiveness of objectives and constraints to linear forms. The graph formalism in Eq 1, however, remains valid even for complex, non-linear constraints and objectives. In this case, more flexible optimization methods have to be used and optimality guarantees might be lost.

The framework we introduced mainly focuses on improving the selection and assembly of epitopes included in a vaccine. However, several other important issues have to be considered when design successful vaccines. The delivery method of such a vaccine is of prime importance, and several different technologies, such as nanoparticles [53], viral vectors [54], and dendritic cell-based techniques [55] are under active research. Viral escape and immunosuppression mechanisms should also be taken into account, as well as the need to generate a heterogeneous response that activates cytotoxic T cells, helper T cells, and B cells. The considerable differences among human hosts and pathogens, the many unknown factors that govern diseases, and the limited understanding of how the components of the immune system

contribute to immunity make it impossible, at present, to create a comprehensive and generally applicable solution for end-to-end vaccine design [56].

The large number of unknowns and the unique features of each disease also affected our simplistic choice of immunogenicity function. Ideally, the immunogenicity should indicate activation of T and/or B cells as measured by the production of cytokines and antibodies, but these events cannot be predicted accurately with current computational approaches, hence our choice of binding affinity as a proxy. We showed that the results are not overly sensitive to the choice of binding affinity predictor, and indeed our framework is completely agnostic to the specific meaning of immunogenicity and edge weight, remaining applicable even when more sophisticated predictors become available. In particular, other immunogenicity objective functions introduced in the context of rational vaccine design [13–15, 18] could also be used.

We assumed that epitopes contribute independently to the immunogenicity of the vaccine, but this is not always verified in practice because of a phenomenon called immunodominance [57, 58]. The presence of certain epitopes is known to reduce or eliminate the response to other epitopes. Surprisingly, it is also possible for an epitope to dominate a stronger binding one so that the immune response is directed towards the weaker binding, but dominating, epitope. Because of the many factors involved, immunodomination is poorly understood and no published predictor is available, hence our linearity assumption. When this task becomes feasible, Eq 2 can be modified to include a linear immunodomination predictor or pre-computed $n$-ary interactions, although the latter approach would be computationally practical only for a small number of epitopes.

To conclude, the proposed framework unifies all commonly used EV design principles, while being agnostic to the specific predictors for immunogenicity and cleavage, as well as the method employed to solve the optimization problem. It enables the design of correctly cleaved string-of-beads vaccines with the largest possible immunogenicity under this constraint. It also enables the exploration of the Pareto frontier between these two competing properties to find the optimal design that best reflects the envisioned trade-off. At the same time, our framework can be used to decompose long vaccines in shorter polypeptides that, together, maintain the properties of the longer sequence. This makes the resulting vaccine easier, cheaper, and quicker to synthesize. We showed that conservation should be emphasized over coverage for mosaic vaccines and demonstrated that the conclusions drawn here are robust with respect to different immunogenicity predictors and the subset of pathogenic sequences used to extract the epitopes.

## Materials and methods

### Data and preprocessing

**Dataset.** We searched whole-genome HIV-1 subtypes B and C sequences in the Los Alamos HIV database [59, 60] collected after 2003, which resulted in 2,241 sequences at the time of writing. These filters were used to ensure higher quality sequences collected with recent technologies. We chose to focus on subtypes B and C as they are the most prevalent across the Americas and Europe (subtype B), and Asia (both subtypes). Since we only included two subtypes, we cannot estimate how effective a vaccine designed in this setting would be on patients infected with a different subtype of HIV-1. An investigation on the input sequences needed to create a truly global HIV vaccine, if at all possible, is left to the practitioner.

We then extracted the Nef gene to limit the computational requirements, finding 1,917 unique sequences, which were used to create five bootstraps, each composed of 300 randomly selected sequences (with replacement). This was done for computational ease, as well as to

study the variability and the generalizability of the results. S1 Data contains the 2,241 Nef genes, while S2 Data contains the sequences used in each bootstrap.

**MHC alleles.** We used 27 MHC alleles and their frequencies found in Toussaint *et al.* [16], reproduced in S1 Table, which together provide a maximum theoretical coverage of 91.3% of the world population.

**Epitopes.** We considered all possible substrings of nine characters as potential epitopes. The binding affinities between peptides and MHC alleles were predicted with NetMHCpan [25], PickPocket [26], or MHCflurry [27], depending on the experiment. S3 Data contains the immunogenicity of each epitope as defined in Eq 2 and predicted by NetMHCpan, since we used that in all but one experiments, as well as the MHC alleles to which it binds and the index of the sequences that contain it.

**Cleavage likelihood.** We computed the cleavage likelihoods between all pairs of epitopes in two ways. We considered the epitopes to be joined directly and used a linear model based on position-specific scoring matrices to predict the cleavage likelihood. We also joined epitopes with optimized spacer sequences, designed to further increase the likelihood of favorable cleavage.

When epitopes are joined directly, we used PCM matrices [29] to compute the weight. The cleavage likelihood at the $k$-th position of a sequence $s$ with respect to an unspecified prior cleavage likelihood $p(C_k)$ is computed as follows:

$$\phi_C(s, k) = \sum_{i=-4}^{1} \psi(s_{k+i}, i) = \log \frac{P(C_k | s_{k-4}, \dots, s_{k+1})}{p(C_k)} \tag{4}$$

where $C_k$ is the cleavage event at position $k$, $\psi(a, i)$ is the PCM matrix, indicating the contribution of amino acid $a$ at offset $i$ with respect to the cleavage site. A positive score indicates that cleavage is more likely than the prior cleavage probability $p(C_k)$, and is usually assumed to indicate a cleavage site. We now compute the negative cleavage likelihood of a sequence composed of two joined epitopes $e_i$ and $e_j$, with $e_i$ of length $\ell$ by adding to the negative cleavage score at the correct position $\ell$ (i.e., between $e_i$ and $e_j$) the cleavage scores at $K$ wrong positions around $\ell$, weighted by a factor $0 \leq \beta \leq 1$:

$$w(e_{ij}) = -\phi_C(e_i e_j, \ell) + \beta \sum_{k=1}^{K} (\phi_C(e_i e_j, \ell - k) + \phi_C(e_i e_j, \ell + k)) \tag{5}$$

where $\phi_C(s, i)$ is the cleavage score at position $i$ of the sequence $s$. We used $K = 2$ and $\beta = 0.1$.

We also applied Schubert & Kohlbacher's framework [17] to design optimal spacers joining all pairs of epitopes and used the optimal objective found by this framework as edge weight in our graph. In this setting, a string-of-beads designed with our framework includes optimal spacers joining epitopes. The spacers were designed by optimizing the cleavage likelihood at the junction between epitopes and spacers, while at the same time minimizing the immunogenicity of possible artificial epitopes that would be formed by incorrect cleavage. We consider values between length zero and four as possible spacer lengths and choose the length with the largest objective value. Note that this implies that some epitopes may be best joined directly without any spacer at all.

We have also been working on an improved framework [61] to design string-of-beads vaccines with optimal spacers that approaches the two problems together and, unlike the present framework, allows residue-level control of cleavage probabilities. It further uses Monte Carlo cleavage simulations to provide a more realistic evaluation of the vaccines' effectiveness.

**Implementation.** The framework was implemented in Python [62], using Pyomo [63, 64] to formulate the linear program and Gurobi [65] to solve it. FRED2 [66] was used to provide

access to bioinformatics tools such as OptiTope [13], [25], PickPocket [26], MHCflurry [27] and PCM [29]. The results were analyzed and visualized in the IPython environment [67], with the aid of NumPy [68], Scipy [69], Pandas [70], statsmodels [71], Matplotlib [72] and Seaborn [73].

## Evaluation metrics

Given a set of epitopes $P$ comprising the vaccine, we compute the following metrics, in addition to the immunogenicity $I(P)$:

**Population coverage.** Given a set of MHC alleles $A$, we define the population coverage of $P$ as the probability that a person has at least one MHC allele binding one or more epitopes of $P$ [16]:

$$p_P = 1 - \prod_{i=1}^{m} \left( 1 - \sum_{a \in A_i} y_a p_a \right)^2 \tag{6}$$

where $A_i$ is the set of alleles of locus $i$, $p_a$ is the probability that a person has this allele, and $y_a$ is a binary variable indicating whether $P$ contains an epitope binding to $a$. Binding was determined by an $IC_{50}$ affinity of at most 500 nM. Note that the graph can contain some epitopes that do not bind to any MHC allele.

According to Eq 6, the coverage of the 27 alleles we chose is 93.1%. In the results, therefore, we report the population coverage relative to this maximum, so that 100% relative coverage corresponds to 93.1% actual coverage.

**Pathogen coverage.** The number of distinct pathogen sequences that contain at least one epitope of $P$.

**Conservation.** The average of the conservation of each epitope in $P$. The conservation of an epitope is the number of proteins that contain it. High conservation indicates a low mutation rate, hence important for the correct functioning of the pathogen.

## Supporting information

**S1 Appendix. Additional methods.** Section A contains a brief description of the $\epsilon$-constrain method [34] to obtain Pareto-efficient solutions in a bi-objective optimization problem, while section B contains the procedure to quantify conserved, low variability pathogen regions in terms of entropy on aligned sequences.
(PDF)

**S2 Appendix. Alternative ILP objectives.** This appendix contains the two alternative formulations of the ILP where average epitope conservation and pathogen coverage are maximized together with immunogenicity.
(PDF)

**S1 Data. Complete set of sequences.** This dataset contains all the 2,241 sequences used in FASTA format.
(FASTA)

**S2 Data. Bootstrapped sequences.** This CSV dataset contains the sequences used in each bootstrap.
(CSV)

**S3 Data. Epitope information.** This CSV dataset contains all the 52,712 epitopes extracted from the sequences, their computed immunogenicity, the MHC alleles that they bind to, and

the index of the sequences where they appear.
(CSV)

**S1 Table. MHC alleles.** This appendix contains a table listing the 27 MHC alleles used in this study and their percent frequency in the world population.
(PDF)

## Acknowledgments

We would like to express our gratitude to Prof. Dr. Bernd Bischl, Dr. Christoph Ogris, and Dr. David Rügamer for their comments on the manuscript.

## Author Contributions

**Conceptualization:** Emilio Dorigatti, Benjamin Schubert.

**Data curation:** Emilio Dorigatti.

**Investigation:** Emilio Dorigatti, Benjamin Schubert.

**Methodology:** Emilio Dorigatti, Benjamin Schubert.

**Project administration:** Benjamin Schubert.

**Resources:** Emilio Dorigatti.

**Software:** Emilio Dorigatti.

**Supervision:** Benjamin Schubert.

**Validation:** Emilio Dorigatti.

**Visualization:** Emilio Dorigatti.

**Writing – original draft:** Benjamin Schubert.

**Writing – review & editing:** Emilio Dorigatti, Benjamin Schubert.

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
