## [Decision Letter · Decision Letter 0]

29 May 2020

Dear Mr Dorigatti,

Thank you very much for submitting your manuscript "Graph-theoretical formulation of the generalized epitope-based vaccine design problem" for consideration at PLOS Computational Biology.

As with all papers reviewed by the journal, your manuscript was reviewed by members of the editorial board and by several independent reviewers. In light of the reviews (below this email), we would like to invite the resubmission of a significantly-revised version that takes into account the reviewers' comments.

We cannot make any decision about publication until we have seen the revised manuscript and your response to the reviewers' comments. Your revised manuscript is also likely to be sent to reviewers for further evaluation.

Sincerely,

Roger Dimitri Kouyos

Associate Editor

PLOS Computational Biology

Rob De Boer

Deputy Editor

PLOS Computational Biology

Reviewer's Responses to Questions

**Comments to the Authors:**

Reviewer #1: Comments uploaded as attachments

Reviewer #2: The authors (Dorigatti and Schubert) describe a novel graph-theoretic algorithm using integer linear program in order to evaluate design principles T-cell epitope vaccines. They distinguish between three types of epitope assembly: epitope mixture, string-of-beads, and mosaic vaccines and provide information on the trade-off between optimal epitope selection and optimal epitope assembly. Further they could show advantage of mosaic over string-of-beads beads and give recommendations on effective design for mosaic vaccines including information on conservation, pathogen coverage, immunogenicity, and population coverage.

The advantage of this method is that the complete design process is covered and a very sound mathematical framework with reasonable definition of constraints were used and that linear programming solvers guarantee convergence and optimality. The manuscript is well written and the method/framework is clearly presented.

Major issues:

1) The authors should discuss and guide the reader a bit more through the differences (or common features and applicability) of the presented model to other vaccine types such as the personalized therapeutic cancer vaccines or vaccines triggering the humoral immune response to produce antibodies.

2) The authors define the vaccine immunogenicity function vaguely as the ability to induce a broad immunization in a target population using sum of products from HLA allele frequencies

times individual immunogenicity as predicted from NetMHCpan. For NetMHCpan was suggested to use ‘% rank’ which is compared to a background set rather than ‘IC50’ as it is better comparable between different HLA types. The authors may provide more evidence that the effect of changing individual immunogenicity (MHC binding affinity) prediction using e.g. other prediction tools could lead to similar results and conclusions.

3) For the string-of-bead vaccines the question of spacers should be mentioned and discussed (see publication on BioRxiv by the same authors https://doi.org/10.1101/2020.04.25.060988)

4) Further should be studied and discussed whether the order of the epitopes have some effect on the results and conclusion.

Minor issues:

Which solver were used for ILP?

Line 65 poylpeptide=> polypeptide

Figure 4 color codes from the legend does not fit (white?)

Figure 5 missing information on y-axes

Line 130 missing end of header

**Have all data underlying the figures and results presented in the manuscript been provided?**

Reviewer #1: Yes

Reviewer #2: Yes

PLOS authors have the option to publish the peer review history of their article (what does this mean?). If published, this will include your full peer review and any attached files.

Reviewer #1: No

Reviewer #2: No
---

## [Decision Letter · Decision Letter 1]

11 Aug 2020

Dear Mr Dorigatti,

We are pleased to inform you that your manuscript 'Graph-theoretical formulation of the generalized epitope-based vaccine design problem' has been provisionally accepted for publication in PLOS Computational Biology.

Best regards,

Roger Dimitri Kouyos

Associate Editor

PLOS Computational Biology

Rob De Boer

Deputy Editor

PLOS Computational Biology

Reviewer's Responses to Questions

**Comments to the Authors:**

Reviewer #1: The authors significantly improved the manuscript and thoroughly addressed all my major concerns. It now reads much better. Some minor comments below, mainly typos.

Minor:

1. Author summary, first sentence: those diseases should be plural.

2. 13: professional cells!?

3. Fig2., legend: needs to be: the edge weight represents or the edge weights represent

4. 196: optimality instead of optimally?

5. 266: the sentence “The string-of-beads constructs contained ten epitopes” seems out of place. 6. 343: the word epitopes is needed here

7. 508: had not having

Reviewer #2: The authors have adequately and comprehensively addressed all raised issues and performed all comparisons with other MHC binding prediction tools and settings and considered the order of epitopes and how the presence of certain epitopes can inhibit or suppress the response.

**Have all data underlying the figures and results presented in the manuscript been provided?**

Reviewer #1: Yes

Reviewer #2: Yes

PLOS authors have the option to publish the peer review history of their article (what does this mean?). If published, this will include your full peer review and any attached files.

Reviewer #1: No

Reviewer #2: **Yes: **Hubert Hackl

---

## [Editor Report · Acceptance letter]

14 Oct 2020

PCOMPBIOL-D-19-02158R1 

Graph-theoretical formulation of the generalized epitope-based vaccine design problem

Dear Dr Dorigatti,

I am pleased to inform you that your manuscript has been formally accepted for publication in PLOS Computational Biology. Your manuscript is now with our production department and you will be notified of the publication date in due course.

With kind regards,

Laura Mallard
